# Tailoring Heat Transfer and Bactericidal Response in Multifunctional Cotton Composites

**DOI:** 10.3390/nano13030463

**Published:** 2023-01-23

**Authors:** Lilian Pérez Delgado, Adriana Paola Franco-Bacca, Fernando Cervantes-Alvarez, Elizabeth Ortiz-Vazquez, Jesús Manuel Ramon-Sierra, Victor Rejon, María Leopoldina Aguirre-Macedo, Juan José Alvarado-Gil, Geonel Rodríguez-Gattorno

**Affiliations:** 1Merida Unit, Functional Materials Laboratory, Applied Physics Department, Center for Research and Advanced Studies (CINVESTAV), Merida C.P. 97310, Mexico; 2Merida Unit, Laboratory of Applied and Molecular Microbiology, National Technological Institute of Mexico, Merida C.P. 97118, Mexico; 3Merida Unit, Aquatic Pathology Laboratory, Marine Resources Department, Center for Research and Advanced Studies (CINVESTAV), Merida C.P. 97310, Mexico

**Keywords:** silver nanoparticles, cotton fabrics, photo-thermal conversion, bactericidal response

## Abstract

Through the execution of scientific innovations, “smart materials” are shaping the future of technology by interacting and responding to changes in our environment. To make this a successful reality, proper component selection, synthesis procedures, and functional active agents must converge in practical and resource-efficient procedures to lay the foundations for a profitable and sustainable industry. Here we show how the reaction time, temperature, and surface stabilizer concentration impact the most promising functional properties in a cotton-based fabric coated with silver nanoparticles (AgNPs@cotton), i.e., the thermal and bactericidal response. The coating quality was characterized and linked to the selected synthesis parameters and correlated by a parallel description of “proof of concept” experiments for the differential heat transfer (conversion and dissipation properties) and the bactericidal response tested against reference bacteria and natural bacterial populations (from a beach, cenote, and swamp of the Yucatan Peninsula). The quantification of functional responses allowed us to establish the relationship between (i) the size and shape of the AgNPs, (ii) the collective response of their agglomerates, and (iii) the thermal barrier role of a surface modifier as PVP. The procedures and evaluations in this work enable a spectrum of synthesis coordinates that facilitate the formulation of application-modulated fabrics, with grounded examples reflected in “smart packaging”, “smart clothing”, and “smart dressing”.

## 1. Introduction

In recent years, innovations in biomaterials have made a tremendous impact in all fields of medicine [1,2]. Nanotechnology has served in this regard as an excellent platform for the next generation of materials, giving inert materials the ability to interact with their environment and respond actively [3,4]. Known as “smart biomaterials”, these multifunctional composites can be engineered to respond to biological, chemical, and physical signals, including pH, redox potential, enzyme activity, temperature, humidity, light, sound, and stress [2]. Rising interest in these materials is driven by their increasing value in precision care and their adaptability to individual needs, from monitoring vital signs, thermal management, moisture management, UV protection, and antibacterial activity to self-cleaning/decontamination capabilities [1,2].

Finishing fabrics made from natural and synthetic fibers have been a major focus in textile manufacturing to achieve a desirable feel, surface texture, color, and ultimately special functional properties. Capable of skin heat transfer with minimal energy input [5], rather than relying on central cooling systems, responsive textiles are emerging as an effective and energy-efficient example of achieving human body thermal comfort or protection [6], moving into the age of “smart clothing”. The development of nanofunctional fibers has also been directed, for example, to the manufacture of hygienic fabrics for odorless products, such as socks, stockings, and underwear [7].

Traditional wound dressing products, such as gauze pads, plasters, and bandages, are comprised of woven and nonwoven fibers of cotton, rayon, polypropylene, and polyester, which can also be functionalized to meet the expectations of smart textiles [8]. “Smart wound dressings” that can interact with wounds, sense, and react to the wound condition or environment, have been proposed to facilitate wound healing effectively [9,10,11]. The dynamic and complex mechanism of wound healing includes stages of coagulation and homeostasis, inflammation, proliferation, and maturation; in chronic wounds, this is a deregulated process that involves infections in the damaged area and increased body temperature [12]. Thermosensitive wound dressings with antimicrobial properties offer a solution to these problems, in which cellulose provides an ideal alternative as a substrate.

Cellulose is the most abundant natural fiber on the planet. It has unique biocompatibility characteristics, porous morphology, and abundant reactive hydroxyl functional groups [13,14], making it an ideal platform for functional materials. In composite materials, noble metal nanocrystals such as silver (Ag) are one of the fillers of excellence to improve the polymeric matrix [15], with special optical and electronic properties that can be controlled from the synthesis by parameters that influence the shape and size of the nanoparticles (NPs) [16,17,18]. It is then possible to obtain materials with smart responses using AgNP-coated cellulose matrix prototypes, for which synthetic routes that guarantee the formation and adherence of the NPs are required. In this scenario, the “polyol process”, a term coined by Fiévet, Lagier, and Figlarz [16], is positioned as a powerful, scalable, and reliable wet chemical route to develop metallic NPs defined in size, shape, composition, and crystallinity, using polyalcohols that perform dually as solvents and reducing agents [19]. The process can be supported by surface stabilizing agents such as polyvinylpyrrolidone (PVP) for fine control of colloidal growth, where the molar ratio with silver nitrate (AgNO_3_) modulates the dimensions and morphologies of the AgNP [20,21,22].

The current scientific literature agglomerates several reports dedicated to cellulosic materials coated with AgNPs. For example, Wu et al. report the development of a superoxide anion monitoring sensor using AgNPs@cellulose as a platform; however, since no correlations between material response and synthesis parameters are established, the chemical and physical interactions that improve $O_{2}^{•-}$ detection are not resolved [23]. Atta and Abomelka explored multiple response properties of Ag-functionalized cellulose (antibacterial activity, electrical conductivity, superhydrophobicity, catalytic activity, ultraviolet blocking, and staining properties) but overlooked comparative studies that could reveal the phenomena behind each response and how it could be modulated [24]. Nam and co-workers delved into the exceptional heat transfer properties of AgNPs@cellulose composites, but with only two samples analyzed, the possibility of providing information on how the AgNP formation and coating process influences the desired thermal response stages is frustrated [25]. Despite the undeniable interest in smart cellulose textile materials, scientific reports focus on specific results that lack comparative analysis schemes, making it impossible to define a guideline for obtaining pre-designed and pre-targeted smart responses. The large-scale production of smart biomaterials is and will be conditioned by the discovery of reliable, sustainable, and efficient synthesis methodologies.

The expansion of smart materials science will be held back as long as procedures that polish scientifically proven materials are not addressed. In order to correlate the elaboration procedure of smart cellulosic materials with their functional properties, an exploratory and exhaustive synthesis analysis is necessary but currently absent. The present work designs a complete processing methodology for AgNP-coated cotton-based smart fabrics with simultaneous thermal and bactericidal response properties. The procedure involves a multiple-synthesis device and parameter configurations, which aim to find the ideal conditions for the maximum expression of a “smart” response in AgNP@cellulose composites. Combinations of temperature, reaction time, and surface modifier concentration enable the proper synthesis coordinates to drive chemical and physical properties, which regulate the material’s functional properties. Heat transfer experiments with heating and cooling stages descriptions were performed and correlated with the differences in the synthesis conditions of each material. Our work also seeks to answer questions about the role that PVP plays in textile coating processes. At the same time, an antibacterial confrontation test with populations of bacteria obtained from three natural water deposits (beach, cenote, and swamp) of the Yucatan Peninsula was carried out as a proof of concept that goes much further than the classic double kinetic evaluation between fundamental bacterial groups (Gram-positive and Gram-negative, also addressed here).

## 2. Materials and Methods

### 2.1. Materials

For the synthesis procedure, silver nitrate (AgNO_3_, ACS reagent 99%; Sigma-Aldrich, San Luis, MO, USA) and polyvinylpyrrolidone (PVP, Mw~55,000; Sigma-Aldrich, San Luis, MO, USA) were used. Desized, scoured, and bleached bare-woven cotton fabric was obtained from a local trader specializing in textiles and haberdashery (ASSIS, Mérida, Mexico) and was used as substrate. For later use, the fabric was fragmented into equal size pieces (1.5 cm × 1.5 cm) and, to ensure the removal of residual chemicals, was subsequently washed with water and laboratory-grade detergent (Alconox^®^, Powdered Precision Cleaner; Alconox Inc., White Plains, NY, USA) at 40 °C and 100 rpm, for 24 h. As a last step, continuous washes were carried out with water, isopropanol, and acetone. The fragments were dried in a vacuum oven (Isotemp^®^ Model 280A; Thermo Fisher Scientific Inc., Waltham, MA, USA) at 60 °C for 24 h. The water used in all the experiments was purified using a WaterPro PS Polishing Station (18 megohm/cm, Labconco Corp., Kansas City, MO, USA). Chemical substances were used without further purification.

### 2.2. Functionalization and Design of “Smart” Cotton Fabrics

The general coating methodology (Appendix A) involved the design and fabrication of a multiple-synthesis device in order to guarantee simultaneous, controlled, and reliable reactions. For this procedure, a device composed of a resistant glass container (Pyrex^®^) and a metal lid was designed. The metallic cover was manufactured with six hexagonally distributed holes and a central hole for placing a temperature sensor and vapor outlet. The glass container was used as a silicone bath and heated on a grill at a stable temperature of 100 °C, 130 °C, and 160 °C, respectively. On the metal cover, six vials with caps were placed, each containing 10 mL of ethylene glycol. The vials were kept semi-open under these conditions for 1 h at 150 rpm, to eliminate possible aqueous residues. Subsequently, a segment of previously cut and washed textile material was added to each vial and kept under the same agitation and temperature for an additional 30 min. The experimental approach was designed for three sets of simultaneous synthesis reactions and their respective repetitions.

#### 2.2.1. Immobilization In Situ of AgNPs

For the synthesis process, two solutions were prepared, one containing 0.500 g of PVP and the other 0.508 g of AgNO_3_; both were dissolved in 25 mL of ethylene glycol. To start the reaction, volumes of 350 µL and 750 µL of PVP solution were added, respectively, into the vials sets one and two, but the third set did not receive this mixture. Immediately, 350 µL of silver nitrate solution was added to all sets of vials, which were hermetically sealed and left to react for 30 or 60 min. This experimental scheme allowed evaluation of the influence of the reaction parameters on the in situ coating of AgNPs on the textile fibers, considering three temperature values, two reaction time values, and the presence or absence of the PVP as a surface modifier agent (PVPx0, PVPx1, and PVPx2). The values of agitation and concentration of silver reagent were kept constant in all experiments. Reactions were stopped by removing the vials from the device and placing them in a container with room-temperature water.

Textile-coated material was subtracted from each vial with a plastic clamp, and reaction residues were removed with continuous washes of water, isopropanol, and acetone. Subsequently, they were dried in a vacuum oven (Isotemp^®^ 280A; Thermo Fisher Scientific Inc., Waltham, MA, USA) at 60 °C for 24 h.

#### 2.2.2. Characterization of AgNP-Coated Cotton Fabrics

Coated cellulose fibers were studied by a field-emission scanning electron microscope (SEM, Model JSM-7600F; JEOL USA Inc., Peabody, MA, USA). The synthesis products were characterized using X-ray diffraction (XRD, D8 ADVANCE with Cu Kα radiation and Ni filter; Bruker Corp., Billerica, MA, USA). The Raman spectra were obtained using a system with excitation and collection from above in a DXR2 model from Thermo Fisher Scientific Inc. (Waltham, MA, USA); the objective lens conditions were 10× with a resolution of 2.7–4.2 cm^−1^ using an excitation wavelength of 532 nm. UV–vis absorbance spectroscopy was measured from 400 to 900 nm using a spectrometer (AvaSpec-ULS2048CL-EVO, Avantes B.V., Apeldoorn, Netherland) with a tungsten-halogen lamp and an integrating sphere (Ocean Optics Model ISP-50-8-R-GT, spectral range 200–2500 nm; Ocean Insight, Orlando, FL, USA). Thermogravimetric curves were measured in a high-resolution thermobalance equipment (Model Discovery 5500, TA Instruments, New Castle, DE, USA). Color coordinates were determined by color measurement software (AvaSoft 8.0; Avantes B.V., Apeldoorn, Netherland) using the three coordinates (L*, a*, and b*) of the CIELAB color system at D65/10°.

#### 2.2.3. AgNP-Coated Release Kinetics

The detachment of AgNPs from cotton fibers was monitored by placing each sample in a suitable plastic container with 50 mL of deionized water at 37 °C and 150 rpm. The aliquots were taken on days 1, 3, 5, and 10. The AgNPs contained in the samples were extracted using a microwave acid digestion system and quantified using a trace elemental inductively coupled plasma mass spectrometry analyzer (ICP-MS, model iCAP^®^; Thermo Fisher Scientific Inc., Waltham, MA, USA). An indium solution was used as an internal standard during the analysis to quantify the recovery process and assay performance.

### 2.3. Temperature Response Measurements

The material’s capacity to dissipate heat was evaluated by following the evolution of the temperature on the surface of the samples using two methodologies. In both cases, the radiation emitted by a flashlight lamp with variable focus and fan-cooled light unit with Task resolved11″ reflector (Model 206, 4800 watts; Speedotron Corp., Bartlett, IL, USA) with an energy of 4.8 kJ per pulse and a duration of 20 ms was used as the heat source. To prevent the emitted infrared radiation on the sample, an infrared filter was placed in front of the lamp. In the first methodology, the analysis was performed remotely using an infrared camera (ImageIR Model 8320, spectral range of 3 to 5 μm; InfraTec GmbH, Dresden, Germany) to obtain a thermogram showing the temporal evolution of the cooling process. A Ge window, transparent to the mid-infrared, allowed the camera lens to be protected from the pulse of light from the flash lamp.

In the second method, a thermistor attached to the back surface of the sample using thermal paste was used as the temperature detector. The evolution of the temperature was followed using a multimeter (Keithley Model 2231A-30-3; Tektronix Inc., Beaverton, OR, USA). The idea of using this method is that even though this is the measurement of only one point of the surface temperature, it is independent of the color and reflectance of the sample, which cannot determine the surface temperature from a thermographic measurement.

### 2.4. Bactericidal Response Measurements

#### 2.4.1. Assay with Natural Samples

The antibacterial activity of the samples was evaluated by combining two techniques: disc diffusion test and time-kill kinetics. The disc diffusion method was carried out considering bacterial populations obtained from natural marine environments (cenote, swamp, and beach).

Before starting this experimental set, the samples obtained from natural environments were freshly cultured in sterile marine media at 37 °C and 100 rpm (the beach sample medium was prepared with NaCl). From 1 mL aliquots, two changes in the marine medium were made to achieve adequate bacterial growth. The inoculum of the third medium change (R = 3) was adjusted in each case to the 0.5 McFarland turbidity standard (absorbance at 625 nm from 0.08 to 0.1, equivalent to 1.5 × 108 CFU/mL) in medium saline (0.85% NaCl). Petri dishes with suitable sterilized nutrient agar were previously prepared and stored for 24 h at 37 °C. From each inoculum, 100 mL was taken and dispersed with a glass spatula onto the plate medium, immediately after the AgNP@cellulose samples (2 mm × 2 mm) were gently pressed against the agar surface. All plates were incubated at 37 ± 2 °C for 18 to 24 h, and the inhibition zones around the samples were visually examined.

#### 2.4.2. Assay with Reference Samples

For the time-kill kinetics assay, reference bacterial cultures (*Escherichia coli* ATCC 25923 and *Staphylococcus aureus* ATCC 25922) adjusted to the 0.5 McFarland turbidity standard (absorbance at 625 nm from 0.08 to 0.1, equivalent to 1.5 × 10^8^ CFU/mL) were used in saline medium (0.85% NaCl). The AgNP@cellulose samples (2 mm × 2 mm) were added to 1 mL bacterial solution, and the reaction was kept under constant stirring at 150 rpm and 37 °C. Aliquots of 100 μL were taken every 1 h to measure the OD600 nm until completing a period of 5 h, and after 24 h the final aliquot was taken. Subsequently, the aliquots were transferred to nutrient agar plates and incubated at 37 °C for 20 h. Viable bacteria were examined through the number of colony-forming units per milliliter (CFU/mL) to monitor their growth. A sample with amoxicillin (1 mg/mL) was used as experiment control (C+), as well as a blank sample without antibacterial material added (C–). The results show the mean values of the three replicates.

## 3. Results and Discussion

### 3.1. Functionalization and Design of “Smart” Cotton Fabrics

#### 3.1.1. Surface Morphology of Coated Fibers

Figure 1 shows selected micrographs of eighteen cotton fiber samples impregnated with silver nanoparticles. The notable contrast in electron density between cotton and metallic silver allows easy identification of nanoparticles and agglomerates dispersed along with the fibers.

Careful inspection of several micrographs of each sample allowed us to discern trends in correlation to the synthesis parameters. The increase in the temperature and time raises the surface coverage, also favoring agglomeration (for example, micrographs A01 and A04, or A07 and A10). The presence and increased concentration of PVP help with particle dispersibility on the surface (for example, micrographs A01 and A03, or A16 and A18). Both behaviors are intuitively expected since increasing the temperature speeds up the rate of nucleation, growth, and agglomeration processes. Increasing the time also favors diffusion and agglomeration of particles at the interface, with a pronounced effect at higher temperatures [26]. In the presence of the PVP polymer, the expected protection against the self-aggregation of the NPs [27,28,29] was evidenced in all the reactions in which this polymer acted.

Typical AgNP synthesis by the polyol method proceeds in four typical steps: (i) dissolution of the metal precursor, which can occur partially at room temperature and be completed during the heating step, (ii) formation of an intermediate phase, which acts as a cation reservoir, (iii) nucleation from Ag monomer species, and (iv) growth leading to the formation of the metallic particles [30,31,32]. In this work, ethyleneglycol was strategically selected as a “polyol” medium for the development of the reaction. This compound, with a large number of available reactive OH groups, offers control over the nucleation, growth, and agglomeration of the particles, providing excellent colloidal stabilization. The simultaneous addition of PVP as a capping agent allows for tailoring the shape and size of NPs by influencing the nucleation and growth steps. Both the oxygen and nitrogen atoms of the pyrrolidone unit promote the adsorption of the long PVP chains on the Ag surface [32], forming a shell-like covering around the cluster and modulating its growth [33]. The synthesis parameters of 160 °C of temperature and 60 min of reaction with a double quantity of stabilizing agent (PVPx2) modulated the formation of smaller and homogeneously distributed NPs on the fibers. Under the same temperature and PVP concentration conditions but with half the reaction time (A15), the behavior of the AgNP coating on the fibers is a reflection that the reaction time was the main parameter for achieving improved dispersibility (A18).

The synthesis parameters of a 130 °C temperature and 60 min of reaction in the presence of PVP modulated the formation of NPs with cubic morphology on the fibers. Silver metal exhibits a face-centered cubic (fcc) structure in which there is no intrinsic driving force for the growth of anisotropic forms [33]. However, it has been suggested that the interaction forces between PVP and different crystallographic facets of an Ag lattice are substantially different and could therefore induce anisotropic growth of this metal (a solid with a highly isotropic structure) [34].

For example, it has been verified that under certain conditions, the selective absorption of PVP in the {100} facets will lead to the preferential addition of silver atoms to the {111} facets with the formation of cubic particles [35]. This line of evidence has never before been observed on cellulose fibers. The synthesis conditions (T = 130 °C, t = 60 min, and PVPx1/PVPx2) exhibited that the presence of PVP modulated the cubic growth of AgNPs, in this case interacting directly with the fibers, which confirms the existence of specific parametric conditions that promote growth to non-typical morphologies.

#### 3.1.2. Surface Chemistry of Coated Fibers

The surface characterization of the composites was evaluated with the support of X-ray diffraction patterns and Raman spectra (Appendix A). The initial bare textile sample (A00) and six AgNP-coated samples (A01, A02, A03, A16, A17, and A18), were selected within the parameter settings for having opposite synthesis conditions. The X-ray diffraction patterns in Appendix A reveal the strongest peaks expected for the crystalline cellulose sample (A00). The group of samples synthesized at 160 °C (A16, A17, and A18), shows a pattern with two additional peaks at 38.31° (Ag {111}) and 44.51° (Ag {200}), which correspond to the main reflections expected for metallic silver [36,37,38]. The zoomed range of the inset in Appendix A reveals the (111) plane reflections. Note that the intensity of the reflections for metallic silver will depend on the degree of coherence, not only on the concentration and particle size, and is also modulated by the volumetric particle distribution and the preferential orientation over the substrate. The low intensity observed for samples at 100 °C seems to be mainly related to lower silver concentration and crystallinity.

The Raman spectra, analyzed in the region from 50 cm^−1^ to 3500 cm^−1^, show different behaviors between the untreated sample and the coated samples (Appendix A). The surface chemistry of A00 reflects a typical spectrum of cellulosic material [37]. In the coated samples, the main Raman signals of cellulose are remarkably enhanced in the “hot spots” that are generated in the nanogaps of plasmonic metal nanoparticles (e.g., Ag) through the amplification of the electromagnetic field caused by localized surface plasmon resonance (LSPR) [38]. The six textile samples coated with AgNPs evaluated by this technique show a similar pattern, with signal amplification observed between approximately 1000 cm^−1^ and 2000 cm^−1^. In all cases, two pronounced peaks are observed at the positions of 1250 cm^−1^ and 1500 cm^−1^, which correspond to assignments of [HCH (twisting), HCC, HOC, COH (rocking) bending], and [HCH scissoring bending], respectively [37]. This result implies the local interaction of AgNPs with these functional groups within the cellulosic structure. The optical images obtained during the measurement of this technique are shown in Appendix A, displaying a visible increase in the material’s pigmentation as the synthesis conditions varied. The darkening of the fibers in the group obtained at 160 °C (A16, A17, and A18) could reflect a greater coverage of AgNPs.

#### 3.1.3. Colorimetric Analysis of Coated Fibers

Figure 2 provides colorimetric information for the eighteen textile samples obtained. Figure 2a arranges photos of the samples according to the distribution of parameters, in order to appreciate the color expressed by the textile under environmental conditions. Color space data, including lightness (L*), red-green (a*), and yellow-blue (b*) were identified and plotted in Figure 2b,c. Establishing a relationship between the different synthesis parameters and the color obtained as a consequence of the differential coating of the AgNPs is an important axis in the emerging smart textiles industry.

In noble metallic particles (e.g., Ag), size-dependent properties are observed, such as surface plasmon resonance (SPR). This optical phenomenon arises from the collective oscillation of conduction electrons in AgNPs when the electrons are perturbed from their equilibrium positions. The frequency and amplitude of the resonance are susceptible to the shape and size of the particles, which determine how free electrons are polarized and distributed on the surface [39]. The sensitive response of surface plasmon peaks can be exploited to optically detect and monitor binding events between the surface of AgNPs and the substrate, in this case, cellulose [40]. This effect translates into a color variation when AgNPs and their aggregates are smaller than 100 nm [41,42,43]. The color has two main effects on how textiles are felt by humans, coolness or warmth; the first is related to the actual physicochemical properties of the material and the other to the sensory perception. Today the textile industry is the world’s largest consumer of dyes, and for these pigmentation processes, a significant amount of water is wasted [42]. For these reasons, synthesis schemes that benefit textile materials in color and intelligence can be industrially attractive.

Figure 2a,b report significant changes in the color of the treated fabrics, mostly related to an increase in the reaction temperature. The simplest approach to engineering the plasmonic properties of an Ag nanostructure is to manipulate its dimensions, the sharpness of corners or edges, and its geometry [44]. The manipulation of thermodynamic magnitudes during a synthesis process, such as temperature, allows for modulating the configuration of the resulting AgNPs [45]. The textile samples synthesized at 100 °C (A01–A06) present a yellowish color with a group luminosity reading (L*) between values of 60 and 80, with minor changes in the a* and b* measurements. These samples, as can be related to the imaging results in Figure 1, show fewer Ag-NP aggregates and better spatially distributed particles. By increasing the reaction temperature to 130 °C (A07–A12) and later to 160 °C (A13–A18), progressive darkness of the samples is observed, with L* values that are concentrated between 40 and 60. The darkening of fibers coated with AgNPs is related to the formation of larger discrete particles and the aggregation of small particles. With an increase in particle size, the absorption band shifts to longer wavelengths. On the other hand, when nanoparticles are close enough to each other, collective interactions between neighboring particles arise. Therefore, the optical absorption of a particular size aggregate comprising smaller particles will be similar to a single large nanoparticle comparable in size to the aggregate [40,44]. In our case, and as Figure 1 confirms, the increase in temperature favors the formation of larger AgNPs and aggregates. The consequence of this effect is a broader absorption spectrum and therefore a change from yellow to brown tones. This tendency is observed to be more pronounced in samples obtained at the highest temperature, where the textile darkening could be the colorimetric result of the collective interactions between AgNPs and their aggregates, with the cellulose substrate. A metallic particle on a substrate will experience an anisotropic environment, where one side of the nanoparticle faces the substrate, and the other side does not [44]; hence, a nanoparticle deposited on a substrate will have different properties compared to the same nanoparticle in a solution. The interaction consequently produces unique tones because of the unusual effects on the plasmonic properties of AgNPs on fibers [42,45], which broadens the spectrum of industrial opportunity for this material.

#### 3.1.4. Thermal Analysis of Coated Fibers

To monitor the thermal stability and composition, thermogravimetric analysis was performed using TGA (Appendix A) from the starting textile sample (A00) and six selected samples coated with AgNPs (A01, A02, A03, A16, A17, and A18). The curves derived from weight loss as a function of temperature were investigated in a range between 25 °C and 850 °C (Appendix A). This result shows two main weight loss events. The first event is similar in all samples, with a gradual decrease in weight between 25 °C and 100 °C, less than 10%, attributed to the evaporation of H_2_O molecules adsorbed on the textile fiber surface. Subsequently, the main mass reduction event occurs between 300–450 °C, which is related to the thermal decomposition of the cellulose polymer [46].

In this temperature range, the uncoated sample (A00) shows a more attenuated descent in weight between 350–450 °C, compared to the samples coated with AgNPs. Similarly, between samples A16–A18, a more pronounced effect is observed in this decay compared to samples A01–A03, which could be an indicator of greater coverage of AgNPs on the fibers (see Appendix A). The presence of metallic particles (e.g., Ag) promotes heat transfer on the substrate surface, thus accentuating the polymeric degradation process in this temperature range. From 500 °C, no weight loss events are observed, maintaining constant values up to 850 °C.

Appendix A summarizes the weight percent values of all samples at 800 °C. The residues of the cotton fabric (sample A00) at 800 °C are 0.95 wt%; this contribution is eliminated in the AgNP-coated samples, and the values shown represent the exclusive percentage weight of the AgNPs. The samples synthesized at 100 °C (A01, A02, and A03) show lower percentage weight values of AgNPs compared to the samples obtained at 160 °C (A16, A17, and A18). This result agrees with the information collected from SEM (Figure 1), XRD (Appendix A), and Colorimetry (Figure 2), showing that in samples A16, A17, and A18 a greater presence of AgNPs is detected. Within the group of samples with less coverage (A01, A02, and A03), a tendency to increase AgNPs is observed when the concentration of PVP increases in the synthesis process. It has been shown that PVP stabilizes the incipient Ag cluster, forming a shell modulating the growth of the nanoparticle. The polymer layer formed is proportional to the reagent concentration, and its stability and degree of coverage are related to synthesis parameters such as temperature and reaction time [31]. The presence of this polymer in A02, and to a greater extent in A03, promotes the formation of AgNPs on the fibers with respect to the sample without this capping agent (A01). On the other hand, in the samples synthesized at 160 °C and 60 min of reaction, the trend is reversed in the presence of PVP. As the temperature rises, the entropy of the system increases, so although the formation of NPs could be favored as the reaction time increases, the polymeric shell stabilizes and probably compacts on the Ag cluster, preventing the correct union of the cluster on cellulose fibers and leaving the largest number of AgNPs in solution.

#### 3.1.5. AgNP-Coated Release Kinetics

In Figure 3, the silver release kinetics from the AgNP@cellulose matrix for 10 days of exposure of six coated samples (A01, A02, A03, A16, A17, and A18) and one bare cellulose sample (A00) are represented. In order to imitate in vitro cell culture conditions, the samples were immersed in an aqueous solution and kept under constant agitation and temperature (37 °C). The amount of Ag detected was determined by inductively coupled plasma mass spectrometry. The concentration released in the deionized water was between 0.24–0.52 ppm in the first 24 h, and from day 5 of immersion until the last day of measurement (day 10), stable release values between 0.57–0.95 ppm are observed. Note that these concentrations exceed the solubility of bulk silver by one order, revealing the nanostructured nature of the system [47].

Figure 3b summarizes the amount of Ag released in the designed experiment by columns, where a greater tendency of release is observed in both groups of samples as the presence of PVP increases in the synthesis process. The group of samples obtained at 100 °C (A01, A02, and A03) releases less Ag compared to the group of samples synthesized at 160 °C (A16, A17, and A18). The analysis of this result demonstrates that the intrinsic characteristics of each material derived from the synthesis process, such as the presence of the polymer or the reaction temperature, have a direct influence on the leakage behavior of the AgNPs adhered to the cellulose textile. In the case of samples A01 and A16, a lower released content is observed with respect to their peers, so it can be inferred that the adherence of silver to the surface is more efficient. As observed in pairs A02 and A17 (obtained with PVPx1) and pairs A03 and A18 (obtained with PVPx2), as the polymer concentration increases, the presence of Ag in the solution also increases, so it is deduced that this reagent induces some steric hindrance in the interaction with the textile, and therefore it is more easily separated. For antibacterial long-term protection purposes, this result is outstanding since a prolonged and constant release of Ag cations greater than 0.1 ppb is needed to effectively inhibit bacterial growth [48,49]. The release values are also within the range stated by the European Food Safety Agency (EFSA) to achieve antibacterial effects, in which the migrated ions must be within the legal limit range of 50 mg Ag^+^/kg of food [50] for food packaging. The release profile behavior of all the samples shows the highest mass fraction release in the first 24 h, which coincides with the generally desired model for drug release. The ability to initially release effective therapeutic concentrations of the active agent and follow a kinetic maintenance behavior are two of the pillars to look for in creating materials with possible medical applications [1], as in the case of smart wound dressings.

### 3.2. Temperature Response Measurements

Figure 4 shows the evolution of the samples’ heat dissipation behavior, summarized in the form of infrared images taken during the first 22 s after the heating pulse. The thermogram shows fast heating of samples and subsequent systematic cooling. The comparative analysis between samples can be observed in a wide range of thermal responses according to the intrinsic characteristics of each evaluated material. For example, sample A18 experiences strong and pronounced changes in temperature, compared to sample A01 with smaller differences in its thermal response over time. However, although the thermogram in Figure 4 allows the heat dissipation process to be visualized, thermographic images can be strongly influenced by the optical properties of the samples and thus are generally not a direct measurement of surface temperature. To perform a numerical analysis of the temporal evolution of temperature in the composites, a second method based on thermistors was used.

The samples’ real temperature profiles are shown in Figure 5, where, in the first stage, the temperature increases in a few seconds, followed by decay at a much lower rate over tens of seconds. The first stage is dominated by the optical absorption of the sample, while the second depends on the emissivity and thermal properties, as well as the interaction with the surrounding environment.

The temperature decrease during the second stage involves several complex processes, in which their functional dependence could be parameterized as multiple time-dependent exponentials [51,52]. However, in these kinds of composite materials, the time interval in which each functional dependence dominates is not easily determined. Therefore, in order to analyze the evolution of the temperature, a practical approach useful in the analysis of decay processes was used [53]. The used criterium consists in determining the decay or cooling time (τ) at which the temperature falls to a third of its maximum value. The results obtained are shown in Figure 6 and Table 1. It is clearly observed in Figure 6 that the cooling times of the samples are divided into two groups. This behavior is related to the optical properties resulting from the AgNPs different agglomeration forms, which is carefully discussed below. The third column in Table 1 shows the concentration of AgNPs fixed to the textile obtained from the thermogravimetric analysis of the samples (see Appendix A). The values of the cooling time (τ) vary from 25 s for A17 to 43 s for A02, and the change of temperature (ΔT) varies from 0.9 °C for A18 to 0.23 °C for A01. Given that the starting textile matrix is the same, the observed differences must be related to the concentration, size, form, and agglomeration of the fixed AgNPs, as well as to the amount of PVP added and the interaction of the compounds as a consequence of the synthesis process. To determine properly the cooling time dependence of each heat transfer requires the use of a more complete and sophisticated experiment, which is in development and will be presented in future work.

The research format chosen in this work included a structured synthesis process, from which samples obtained at 100 °C and 160 °C show clear group trends, regardless of other synthesis parameters. The colorimetric results (see Figure 2) show a behavior in the reported values of L*, a*, and b* that can be grouped according to the synthesis temperature. The thermogravimetric analysis of Appendix A also supports this idea by reporting weight loss behavior as well as percentage Ag concentration values, which are grouped by similarity according to temperature. As previously explained in this section, the absorbance spectra (see Figure 5) also show two types of characteristic profiles between the sample groups at 100 °C and 160 °C. The third of the maximum temperature value (see Figure 6) as well as the integrated absorbance (Table 1) of the samples once again denote a group behavior directed by the same temperature parameter. It is clear through different characterizations that the temperature parameter is key to achieving pronounced changes in material properties. The double plasmon observed in the samples obtained at 160 °C seems to be the consequence of a synthesis process at said temperature value, which is also the leading phenomenon and conditioner of the responses reported by the material.

The results in Figure 5 and the percentage of silver content estimated thermogravimetrically allow an interesting relationship between the analyzed samples. Figure 7 shows the relationship between the thermogravimetrically estimated concentration (wt.%) of AgNPs and the integrated absorbance when the sample undergoes temperature changes after the light pulse [52,53,54]. Opposite trends seem to occur in both groups of samples, apparently related to the AgNP formation process, and the reaction temperature used (100 °C and 160 °C). In the group synthesized at 100 °C greater temperature changes are observed as the silver presence increases, while the 160 °C group experiences higher values of ΔT with a lower AgNP presence. This type of analysis makes it clear that differences in the absorbance of the sample induce differences in the rate of light conversion into heat, but this conversion does not have a strictly linear relationship with the content of silver coating the textile. The trends then seem to respond to a synergy correlation among particle size, concentration, and degree of agglomeration of the AgNPs along the fibers, all affecting the intrinsic optical properties of the material.

By monitoring the samples’ thermal behavior, it is possible to divide the entire process into two stages, where, as mentioned above, the first stage (heating) is closely related to the optical properties of the sample (Figure 5). Samples with high absorbance also show higher values of ΔT, while the ones with low absorbance show smaller values of ΔT. This behavior is related to the absorption effect induced by the LSPR optical phenomenon and provoked by the presence of the silver NPs [55,56]. The coherent oscillations of the electrons in the vicinity of the Ag-dielectric interface strongly depend on the size and form of the particles [57]. On the other hand, numerical and analytical approaches to LSPR behavior in NPs illustrate that when two or more NPs are in proximity, the localized particle plasmons of the individual NPs interact with each other through their near optical fields, creating coupled LSPR modes. The frequency resonances of the coupled NPs therefore depend to a large extent on the distance and configuration between particles [58,59]. The proximity between particles and the degree of agglomeration are determining factors in the plasmonic behavior and therefore direct modifiers of the thermal response of the material. The consequence of the coupling of these oscillations is a spectral shift (in general red-shifts) with respect to the localized particle plasmons of the individual NPs [60,61,62,63]. This in turn is verified through the second band (600–650 nm) observed in the samples of the group synthesized at 160 °C (inset of Figure 5), where it can be inferred that the agglomeration and proximity create plasmonic coupling effects that modulate the differential thermal behavior of the material. The most pronounced effect was observed in sample A18, where the coating of the fibers was visualized with AgNPs that were smaller, closer, and more uniformly scattered than their counterparts that do present agglomerates (A16 and A17) (see Figure 1), stimulating an amplified coupling phenomenon. In this group, the collective interactions of the AgNPs, which also contribute to sample darkening, allow better absorption of light by the material and therefore promote the light-to-heat conversion process. Therefore, the contribution in the conversion of light to heat derived from the LSPR phenomenon in samples A16, A17, and mostly in A18, shows an amplified behavior as a direct consequence of the coupling of the dielectric fields of the NPs and not of the silver content.

During the second stage (cooling), the presence of silver is also decisive. In this case samples A16, A17, and A18 exhibit faster cooling rates (lower than 22 s). The heat dissipation rate into the surrounding air is driven mainly by AgNPs, which have a high thermal effusivity. The thermal effusivity of bulk silver is 35,700 Ws^1/2^/m^2^K, while for a polymer/textile it is around 500 Ws^1/2^/m^2^K [53], seventy times less. Given that thermal effusivity defines the ability of the body to exchange thermal energy with its environment [53], this can explain why the samples with a higher content of silver cool down faster than the others with a lower silver content. The analysis of the samples that do not contain PVP, A01 and A16, the first belonging to the group obtained at 100 °C and the second to the group at 160 °C, respectively, present shorter τ values compared with their group partners. This indicates that the PVP can act as a thermal barrier preventing the sample from cooling down by hindering the flow of heat from the AgNPs, avoiding its prompt dissipation to the environment. On the other hand, when the number of AgNPs increased, heat dissipation was enhanced in each group.

In the first group, the sample that is expected to have the thicker layer of PVP (A03) cools slightly faster than A02. Analogously, A17 cools quicker than A18, with A17 having a higher number of AgNPs. The cooling in these systems comprises a complex process, involving the coupling of the transport of electrons in silver and phonons in the textile and polymer; at the same time, it is affected by the presence of the previously explained plasmonic effects.

### 3.3. Bactericidal Response Measurements

#### 3.3.1. Assay with Natural Samples

Figure 8 shows the bactericidal activity of six AgNP@cellulose composite materials confronted with bacterial populations obtained from the three water reservoirs in the Yucatan Peninsula.

The fresh samples were cultivated in a suitable nutrient medium and were confronted through the agar diffusion test. The textile samples were in all cases effective against bacteria after 24 h in contrast to the control treatment. Figure 8a (marine bacterial population) shows a wide zone of inhibition around the AgNP films, even overlapping with each other. In Figure 8b (freshwater bacterial population), the inhibition halos are even more pronounced, controlling/eliminating bacterial growth in the area where the control textile material (A00) is found. On the other hand, in Figure 8c (swamp brackish water bacterial population) almost a complete annihilation of the bacteria can be observed. Aquatic ecosystems are recognized as some environments with the greatest trophic richness in terms of the organisms that compose them (bacteria, phytoplankton, benthos, etc.). Recent estimates suggest that only 0.1% of existing bacteria have been identified, so the analysis made in this work is based on the knowledge described so far of the microbial fauna of each natural environment. The Gulf of Mexico and the Caribbean Sea are two outstanding marine ecosystems that join in the Yucatán channel. This particular geographical situation favors the existence of a high bacterial diversity such as Bacillus, Brucell, Campylobacter, Carnobacterium, Chlamydia, Edwardsiella, Enterobacter, Legionella, Mycobacterium, Pseudomonas, Salmonella, Serratia, Klebsiella, Lactococcus, Neisseria, Proteus, Staphylococcus, Stenotrophomonas, Streptococcus, Treponema, Vibrio, and Yersinia [64]. The presence of potentially pathogenic bacteria in these coastal ecosystems, mostly Gram-negative bacteria, must be recognized to ensure the safety of the public accessing the beaches. Figure 8a shows the accentuated formation of inhibition halos around each coated textile sample, a response that reflects its bactericidal potential for marine populations.

The Yucatán Peninsula is underlain by extremely permeable limestone that has eroded to form a large network of underground aquifers that are connected through cave passages [65]. The natural collapse or dissolution of water-filled cave ceilings opened the aquifer to the atmosphere, generating thousands of cenotes (sinkholes) over this peninsula [66,67]. This typical karst topography shows two different aquifer ecosystems: open and illuminated water reservoirs (open cenotes) and closed water reservoirs in partial or total darkness (closed or semi-closed cenotes). The microbial community of open cenotes, as in the case of the “Chen-Ha” cenote, has been previously explored with a dominant diversity of genera including Sulfurovum, Sulfurimonas, Methylocystis, Acinetobacter, Methylotenera, Exiguobacterium, Lentimicrobiaceae, Cyanobium, and Gracilibacteria among others in less proportion [68]. Except for the genus Exiguobacterium, the rest of the bacterial community found is concentrated within the Gram-negative group, which suggests again the high sensitivity of this bacterial group (Figure 8b) to the textile composite.

Swamps are considered transition zones because both land and water play a role in creating this environment. In the “Yucalpeten” swamp, some pathogenic bacteria such as the genus Salmonella, Shigella, and Vibrio, as well as indicator type bacteria (of the genus Escherichia and Enterobacter), decomposition type (of the genus Pseudomonas and Flavobacterium), and denitrifying type (of the genus Pseudomonas) have been found, and all these species belonging to the Gram-negative group. The result derived from Figure 8c shows the total sensitivity of the bacterial community that inhabits the swamp to AgNPs, confirming the relationship with the structurally characteristic membrane of Gram-negative bacteria.

#### 3.3.2. Assay with Reference Samples

To evaluate the sample’s antibacterial potential, quantitative survival tests (Figure 9) show the growth of the population of both bacteria, contrasting the bare cotton sample (A00) with the AgNP@cellulose composite materials. In the evaluation with *E. coli*, Gram-negative bacteria, a more pronounced growth decline was found with respect to *S. aureus*, Gram-positive bacteria. In Figure 9a it is observed that from 5 h samples A16, A17, and A18 showed no growth, even after 24 h of the experiment. In the same figure, the bacterial growth in sample A02 is especially retarded compared to its synthesis partners (A01 and A03). On the other hand, in Figure 9b, a weakened growth is observed during the 24 h of the evaluation, where samples A01 and A02 show the smallest slope.

In the AgNP release result (see Figure 3b), the group synthesized at 100 °C (A01, A02, and A03) released a greater number of particles in the first 24 h compared to the group obtained at 160 °C (A16, A17, and A18). Since A02 is a common bactericidal denominator in both assays, the amount of AgNPs released in combination with the presence of the surface stabilizer used for its synthesis (PVP), seems to be the factor that particularly favors its bactericidal capacity. This result is an example that there are specific synthesis conditions that influence the material interaction response, where in this case A02 behaves as a bacteriostatic material that retards growth.

The antibacterial properties of the AgNPs depend mainly on the size, pH, ionic resistance of the medium, and if present, the capping agent. One of the most validated schemes suggests that silver ions (Ag^+^) could be continuously released from Ag–NP and be leading to the antibacterial mechanism of action [69,70]. In addition, as described theoretically by the Ostwald–Freundlich equation [71], the size and shape of the AgNPs influence the production of Ag^+^ ions. According to this, smaller sizes of AgNPs and spherical or quasi-spherical arrangements are more susceptible to the release of Ag^+^ ions due to their larger surface area [72]. Thus, the aggregation of AgNPs reduces the release of Ag^+^ ions. This issue could be resolved with the usage of capping agents, improving the effective dissolution activities of the AgNPs [73].

Figure 9a reveals the growth interruption of *E. coli* in the presence of the group of samples obtained at 160 °C, which also share the property of being the samples with the highest mass of AgNPs covering the fibers (see Appendix A), so it could be intuited that release and death by contact is happening between bacteria and the AgNP@cellulose composites. The electrostatic attraction amongst the negatively charged microbial cells and positively charged AgNPs has been previously observed [62], and these nanostructured systems are proposed as the most apt bactericidal agents against burn wound pathogens [74,75].

The specific mechanism of antibacterial activities or toxicity by AgNP is still indefinite and has not been completely explained. It has been noted that Ag^+^ ions interact with nucleic acids through the phosphate groups of the nucleosides that compose them, causing disruptions in the structure and, therefore, alterations in cell propagation processes [76,77,78]. Due to the electrostatic attractions and affinity towards the sulfur proteins, Ag^+^ ions adhere to the thiol groups of vital enzymes, affecting cellular respiration and transport of ions across membranes, resulting in cell death [79,80,81]. As soon as cells absorb the free Ag^+^ ions, oxidative stress is generated by the formation of reactive oxygen species (ROS) such as oxygen superoxide (O^2−^), deactivating respiratory enzymes, and interrupting adenosine triphosphate (ATP) release [76,82].

Apart from the ability to release the Ag^+^ ions, the AgNPs could themselves eradicate the microbes or bacteria. The negatively charged bacteria have an affinity for positively charged nanoparticles as expected in AgNPs, leading to a strong binding between the nanoparticles and the bacteria surface. The cumulative AgNPs trigger denaturation of the cell membranes and due to nanoscale size, they have the capability to permeate through the cell wall of bacteria and consequently modify the cell membrane arrangements [78,83].

The antibacterial efficacy of AgNPs is also related to the types of pathogenic bacteria, or more specifically to the structures of their outer membranes. Gram-positive bacteria have a thick cell wall composed of multiple layers of peptidoglycans that serve as a barrier for Ag^+^ ions and AgNPs. However, Gram-negative bacteria, with a single layer of peptidoglycans in their structure, are more likely to permeate and therefore more prone to Ag invasion with the consequent damage previously explained [76]. This statement is supported by the results in Figure 8, where it is observed that populations with higher numbers of species of this group are more sensitive to the presence of AgNP@cellulose compounds. In addition, the result of Figure 9 also confirms this analysis, as the growing rate of *E. coli* (Gram-negative) compared to *S. aureus* (Gram-positive) was lower.

## 4. Conclusions

In the present work, an eighteen-strong set of cellulose fabrics impregnated with AgNPs were systematically characterized and the results evidence how the synthesis parameters influence the quality of the coating in terms of particle density and agglomeration, and their impact on the physicochemical properties. Increasing the temperature and the reaction time increases the coverage of NPs, also favoring their agglomeration, while a greater concentration of PVP promotes the dispersibility of the particles, protecting them from self-aggregation. An increase in coverage and aggregation of NPs leads to an increase in the integral absorptance of the composite, allowing it to modulate the color from light yellow to dark brown tones. The strength of the interaction between the AgNPs and the fibers was tested in an aqueous medium, showing that the adherence between the surface Ag^+^ ions and the OH– groups, in the absence of polymer participation (PVP), is stronger and therefore release occurs slowly until equilibrium is found.

The comparative analysis of the photo-thermal conversion and dissipation behavior between samples was demonstrated in visual thermograms and direct measurements of the heating and cooling stages. The differential absorbance of the sample induces differences in the rate of conversion of light into heat with a synergy correlation among particle size, concentration, and agglomeration degree of the AgNPs along the fibers, all affecting the intrinsic optical properties of the material. Therefore, the contribution in the conversion of light to heat derived from the LSPR phenomenon in the group of samples of 160 °C shows an amplified behavior as a direct consequence of the coupling of the dielectric fields of the NPs. With the increase in the coverage of AgNPs and the absence of PVP, greater heat dissipation is observed, confirming that when the polymer is present, it acts as a thermal barrier, preventing the flow of heat and slowing down the cooling of the sample.

The antibacterial capacity was measured in two different confrontation scenarios to cover a greater response spectrum. The kinetic evaluation in an aqueous medium was performed with two standardized bacteria (*E. coli* and *S. aureus*), while the agar diffusion technique was carried out with bacterial populations from three natural water deposits (a beach, cenote, and swamp of the Yucatan peninsula). As expected, in this type of composite, exceptional bactericidal properties were achieved with a more significant elimination impact on bacteria of the Gram-negative group.

We consider that the crucial data obtained in this work manage to explain the differences observed in the thermal and bactericidal attributes, providing the technical and scientific bases to address real-life applications. From handling septic media in hazardous environments to being a temperature stabilizer, these materials can be envisaged as a functional part of wound dressings, containers for food packaging, or antimicrobial and temperature-responsive fabrics. In addition, AgNP immobilization can color textile fibers through the optical effect explained, avoiding the use of toxic agents to fix the dyes. With an impact in the coming years on the medical, food, and textile industries, this multifunctional textile may be part of the solution to problems such as (i) the growing number of multi-resistant bacteria strains, (ii) the increase in the average temperature of the planet without relying on cooling systems, and (iii) the pursuit of sustainable materials to increase shelf life and reduce food waste.

## Figures and Tables

**Figure 1 nanomaterials-13-00463-f001:**
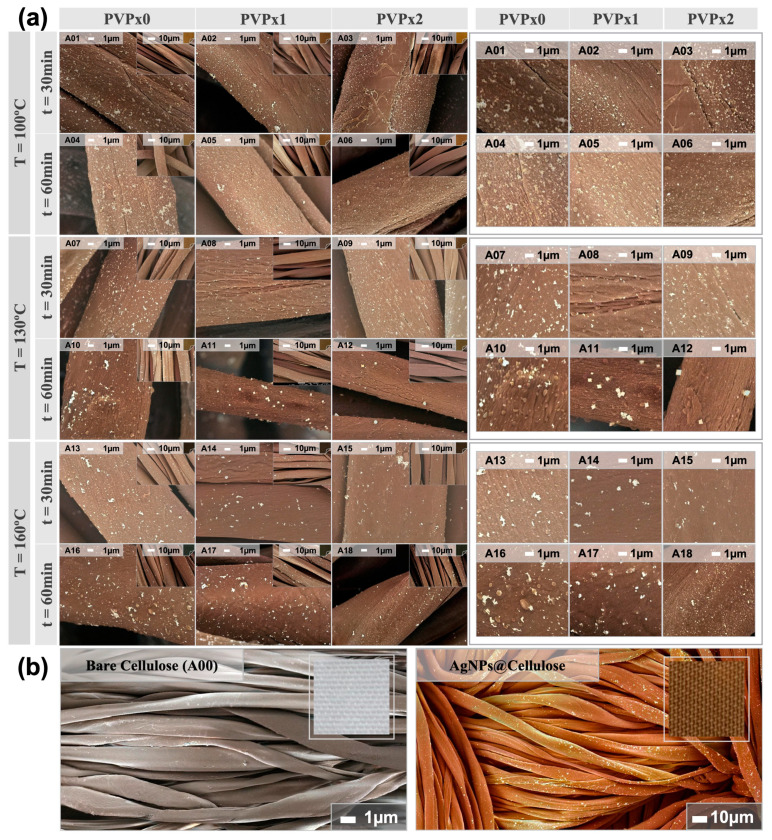
Selected micrographs obtained from SEM studies of textile samples coated with AgNPs and their agglomerates at the cotton surface. The inserts within each image correspond to selected areas zoomed out to visualize the coating on multiple fibers. A set of eighteen textile samples (from A01-A18) coated with AgNPs (**a**) and a bare cotton sample compared to coated sample (AgNP@cellulose) (**b**) are presented. The inset of the comparison between coated and uncoated sample, reveals the real appearance of the textile sample. Samples are classified in the left bar into three temperature parameters (100, 130, and 160 °C) and two reaction time parameters (30 and 60 min). In the upper bar, three PVP concentration parameters are represented (PVPx0 or absent, PVPx1, and PVPx2). For better visual contrast, the color in the images is simulated (false color).

**Figure 2 nanomaterials-13-00463-f002:**
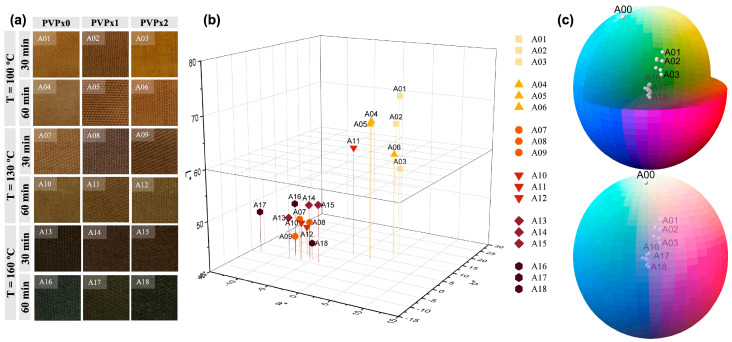
Correlation between real appearance (**a**) and color is represented by a chromaticity diagram (**b**,**c**) of the eighteen textile samples coated with AgNPs.

**Figure 3 nanomaterials-13-00463-f003:**
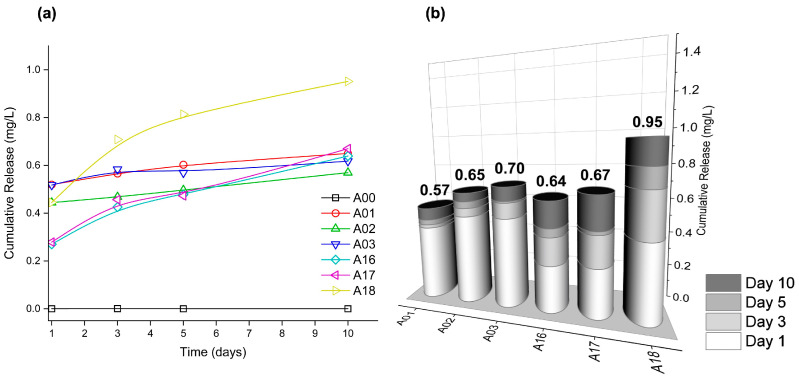
Silver-releasing behavior as a function of the immersion time (**a**) and cumulative silver release (**b**) of bare cotton (A00) and six selected AgNP@cellulose samples: A01 (T = 100 °C, t = 30 min, PVPx0), A02 (T = 100 °C, t = 30 min, PVPx1), A03 (T = 100 °C, t = 30 min, PVPx2), A16 (T = 160 °C, t = 60 min, PVPx0), A17 (T = 160 °C, t = 60 min, PVPx1) and A18 (T = 160 °C, t = 60 min, PVPx2). The cumulative release of Ag in mg/L was obtained from the chemical analysis by inductively coupled plasma (ICP-MS) technique.

**Figure 4 nanomaterials-13-00463-f004:**
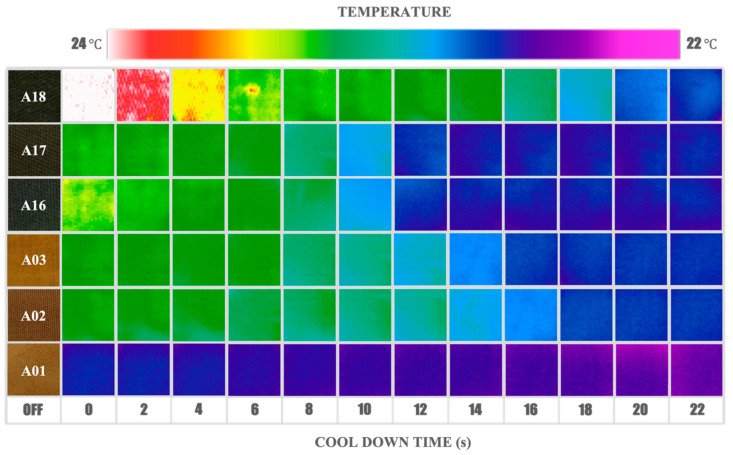
Time dependence of the infrared thermogram of six selected AgNP@cellulose samples: A01 (T = 100 °C, t = 30 min, PVPx0), A02 (T = 100 °C, t = 30 min, PVPx1), A03 (T = 100 °C, t = 30 min, PVPx2), A16 (T = 160 °C, t = 60 min, PVPx0), A17 (T = 160 °C, t = 60 min, PVPx1) and A18 (T = 160 °C, t = 60 min, PVPx2), depicting the first 22 s of cooling process.

**Figure 5 nanomaterials-13-00463-f005:**
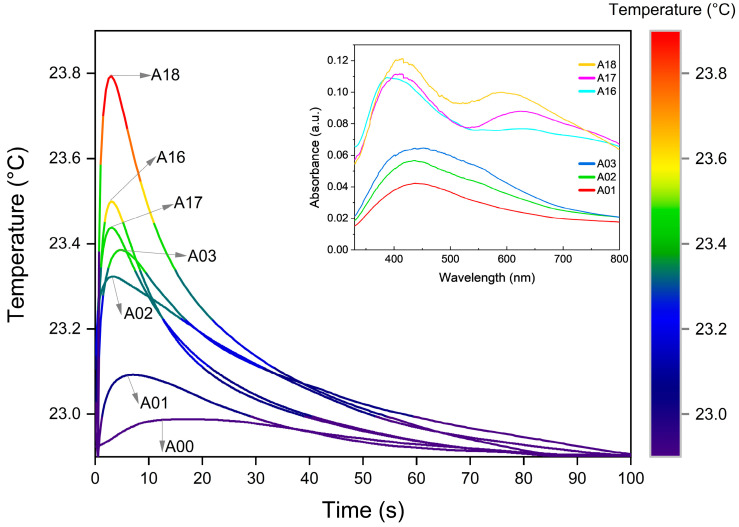
Temperature profile of six selected AgNP@cellulose samples: A01 (T = 100 °C, t = 30 min, PVPx0), A02 (T = 100 °C, t = 30 min, PVPx1), A03 (T = 100 °C, t = 30 min, PVPx2), A16 (T = 160 °C, t = 60 min, PVPx0), A17 (T = 160 °C, t = 60 min, PVPx1) and A18 (T = 160 °C, t = 60 min, PVPx2) heated by the flash lamp measured during 100 s using a thermistor-based technique. In the inset, the corresponding absorbance spectra for the same samples are presented.

**Figure 6 nanomaterials-13-00463-f006:**
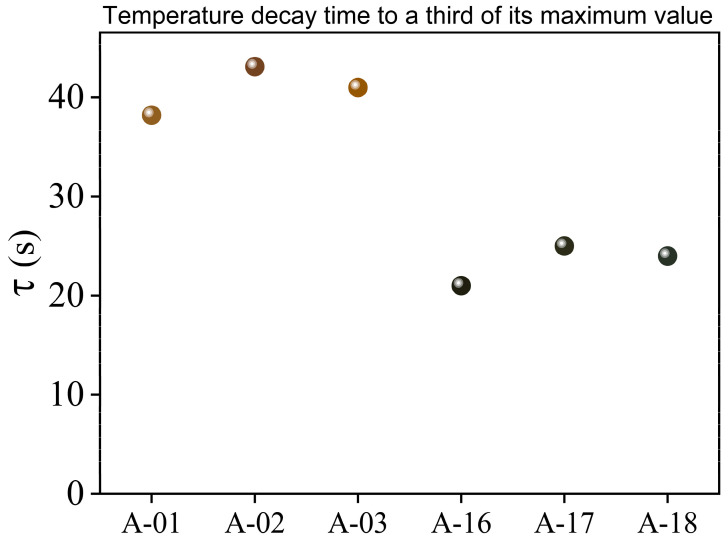
Temperature decay time (τ) to a third of its maximum value of the six selected AgNP@cellulose samples, determined from their temperature profiles shown in Figure 5.

**Figure 7 nanomaterials-13-00463-f007:**
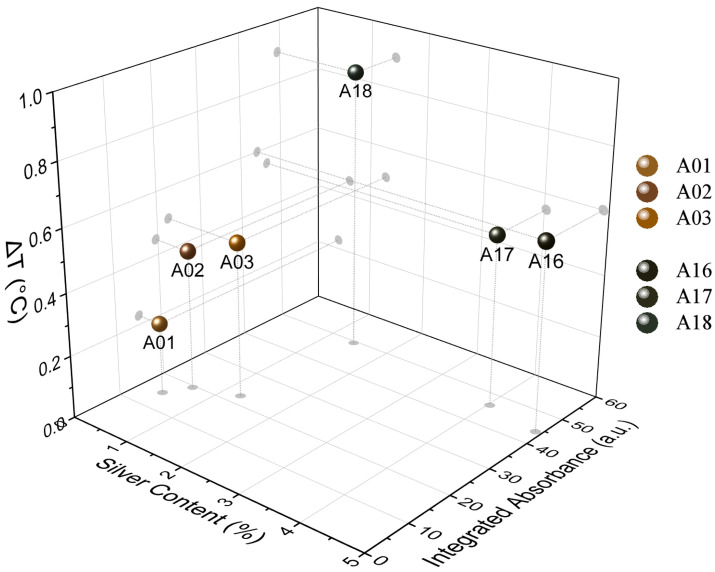
Correlation between the integrated absorbance and the percentage of silver content estimated thermogravimetrically. The relationship is established considering the temperature variations that each sample undergoes after a light pulse. The analysis was performed for the same set of six selected samples of AgNP@cellulose previously analyzed (A01, A02, A03, A16, A17, and A18).

**Figure 8 nanomaterials-13-00463-f008:**
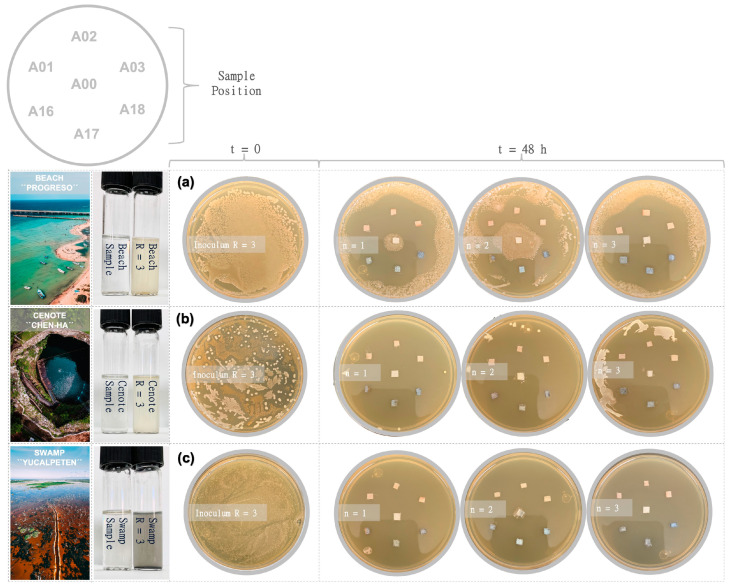
Evaluation of the antimicrobial activity of bare cotton (A00) and six selected samples of AgNP@cellulose (A01, A02, A03, A16, A17, and A18). The samples coated with AgNPs are arranged radially, and in the center is the unprocessed control sample. The confrontation was carried out through the agar diffusion test with bacterial populations obtained from three natural water deposits: Beach (**a**), Cenote (**b**), and Swamp (**c**).

**Figure 9 nanomaterials-13-00463-f009:**
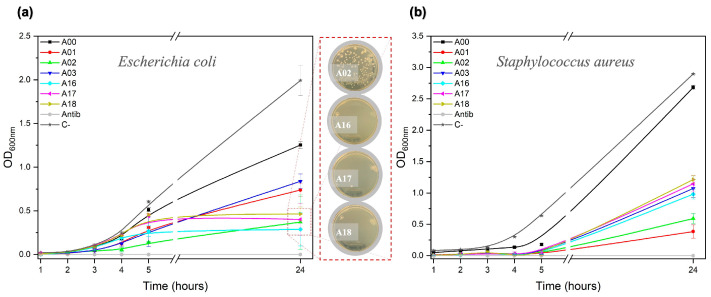
Kinetic evaluation of the antibacterial activity of bare cotton (A00) and six selected samples of AgNP@cellulose (A01, A02, A03, A16, A17, and A18) against *E. coli* (**a**) and *S. aureus* (**b**). The *E. coli* inset represents the bacterial presence after 24 h for samples A00, A02, A16, A17, and A18.

**Table 1 nanomaterials-13-00463-t001:** Parameters extracted from the data analysis: cooling time (τ), percentage of fixed AgNPs obtained from thermogravimetric analysis, and numerical values of the integrated absorbance of the samples are presented.

Sample	τ (s)	AgNPs Fixed (%)	Integrated Absorbance (320–1000 nm)
A01	38	0.40	15.2
A02	43	0.61	19.5
A03	41	1.28	22.5
A16	21	4.91	44.1
A17	25	4.03	46.4
A18	24	1.43	49.4

## Data Availability

Not applicable.

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
