# Peer review of "Tailoring Heat Transfer and Bactericidal Response in Multifunctional Cotton Composites"

_nanomaterials, 2023, doi:10.3390/nano13030463_

Round 1
Reviewer 1 Report
In this manuscript, silver nanoparticles are loaded on cellulose to improve the heat dissipation and antibacterial effect of fabrics to adapt to the needs of smart fabrics. The results showed that it had good antibacterial ability and the heat dissipation effect was greatly improved by controlling the synthesis mode. I think a minor revision should be considered before accepted.
My comments are as follows:
1. There may be other possible reasons for the absence of diffraction peaks of silver particles in the XRD structure of the sample at 100 °C. Some weak diffraction peaks can be revealed by increasing the detection Angle or decreasing the scanning speed.
2. Excluding the influence of plasma, the fabric can be heated to the same temperature and then the temperature change can be observed.
3. In Figure 7, anomalous points appear in the regularity of the 160 °C sample, which can be obtained by enlarging the mass range of silver and refining the mass gradient.
Reviewer 2 Report
This manuscript explored how the reaction time, temperature, and surface stabilizer concentration impact the functional properties in a cotton-based fabric coated with silver nanoparticles (AgNPs@cotton), with respect to thermal and bactericidal response. The authors represented here some important aspects of nanoparticle-polymer conjugates which are very crucial for biomedical applications. This article is thorough and provided a foundation on this matter. However, here is some comments that need to be addressed by the authors in the manuscript before publication:
1. What are the parameters taken into account when the authors selected their textile materials? It would be great if the authors let readers know common types of fabric used for wound dressing etc.
2. There are no TEM pictures of synthesized Ag-nanoparticles. Is there a specific reason?
3. In bacteria killing experiment the unsupported Ag-NP should be included as another control to see the effect of supporting fabric materials.
4. The ratio of Ag0/Ag+ should be measured in these materials. Without this data, it is impossible to clarify the origin of the antibacterial effect.
5. To value the process of “Immobilization in-situ of AgNPs” the authors should include a control sample containing a physical mixture of pre-made Ag-NP and cotton fabrics.
Reviewer 3 Report
This manuscript investigated the “Tailoring Heat Transfer and Bactericidal Response in Multifunctional Cotton Composites”. The subject of this paper is interesting and the results seem to support the enhancement of bactericidal activities. The paper is well written and suitable for publication in nanomaterials journal after the minor revision. The list of commends are shown below.
1. Recent literature comparison in the introduction section is lacking.
2. “In the scientific literature, numerous works explore…….” Add the suitable citations.
3. Many works of literature are available for "smart" response in AgNPs-cellulose composites. However, the detailed literature reports on its advantages and its concerns is not discussed in this manuscript. The author should include this in the introduction section and highlight the novelty of present work.
Round 2
Reviewer 2 Report
The authors have nicely clarified all my concerns and made changes as per suggestions. This manuscript is ready for publication.